# Electrodeposition, Characterization, and Corrosion Behavior of CoCrFeMnNi High-Entropy Alloy Thin Films

Ana-Maria Julieta Popescu [1], Florina Branzoi [1], Ionut Constantin [1,*], Mihai Anastasescu [1], Marian Burada [2], Dumitru Mitrică [2], Ioana Anasiei [2], Mihai-Tudor Olaru [2] and Virgil Constantin [1,*]

1   "Ilie Murgulescu" Institute of Physical Chemistry-IPC, Laboratory of Electrochemistry and Corrosion, 202 Splaiul Independentei, 060021 Bucharest, Romania; popescuamj@yahoo.com (A.-M.J.P.); fbrinzoi@chimfiz.icf.ro (F.B.); manastasescu_ro@yahoo.com (M.A.)
2   National R&D Institute for Nonferrous and Rare Metals-IMNR, 102 Biruinţei Blvd., 077145 Pantelimon, Ilfov County, Romania; mburada@imnr.ro (M.B.); dmitrica@imnr.ro (D.M.); ianasiei@imnr.ro (I.A.); o.mihai@imnr.ro (M.-T.O.)
*   Correspondence: dr_ing_constantin@yahoo.com (I.C.); virgilconstantin@yahoo.com (V.C.)

**Abstract:** Potentiostatic electrodeposition was used to obtain CoCrFeMnNi high-entropy alloy (HEA) thin films on copper substrate. An electrolyte based on a DMSO (dimethyl sulfoxide)-$CH_3CN$ (acetonitrile) organic compound was used for the HEA deposition. The microstructure of the high-entropy deposits before and after corrosion in artificial seawater was investigated by scanning electron microscopy (SEM) and energy dispersive spectrometry (EDS) investigation. SEM analysis revealed that compact and uniform film consists of compact and uniform 50 nm–5 μm particles that form the HEA films. The successful co-deposition of all five elements was highlighted by the energy dispersive spectrometry investigation (EDS). Electrochemical measurements carried out in an aerated artificial seawater solution under ambient conditions demonstrated the promising potential for application in the field of anti-corrosion protection, due to the protective behavior of the HEA thin films.

**Keywords:** electrodeposition; high-entropy alloy; thin films; surface analysis; corrosion protection

## 1. Introduction

HEAs are considered a new class of alloys based on an innovative approach. Unlike traditional alloys, these HEAs are based on equiatomic mixtures of five or more elements and do not have a main component. HEAs microstructure consists of a number of intermetallic compounds according to classical metallurgical theory. Due to large mixing entropy, simple structures with a solid solution or a single-phase crystalline structure can be formed on HEAs. Sluggish diffusion and severe lattice distortions characterize the HEAs. This represents a significant influence on their microstructures and properties. The materials science and engineering community has been greatly attracted to the research of HEA alloys [1–3].

Extensive research conducted to study different HEA alloy systems with special properties (high corrosion and softening resistance, high hardness, wear resistance, good electromagnetic properties, and the ability to maintain their properties at high temperatures) has revealed that these alloys can be applied as functional and structural materials [4–8].

The most common method of preparing HEAs is by using the melting–casting route (arc and induction furnace melting) [9,10]. For obtaining bulk materials, other synthesis processes used are mechanical alloying and rapid solidification [11,12]. Various deposition methods, such as magnetron sputtering and laser cladding, were used for producing HEA coatings [13–17]. Compared with other technologies for the synthesis of HEA thin films with low cost, electrodeposition is considered a good alternative. Electrodeposition is considered a good and cheap alternative because it does not require expensive and complex

equipment and, at the same time, uses raw materials easily accessible. The possibility of obtaining at low processing temperatures and with low energy consumption thin films on substrates with a varied and complex geometry makes this method of electrodeposition to be achieved with simple control of the morphology, chemical composition, and thickness of HEA films by simple variation in the submission parameters.

Due to their special characteristics (good chemical and thermal stability, high electrical conductivity, wide working temperature range, wide electrochemical windows, lack of hydrogen release, and hydroxide generation), non-aqueous electrolytes are considered an optimal alternative for electrodeposition of metals and alloys [18–20].

Few efforts are known in the available literature, until now, for preparing HEAs by electrodeposition. Li et al. [21–24] prepared BiFeCoNiMn, MgMnFeCoNiGd, and TmFeCoNiMn "multi-component HEA thin films by electrodeposition in the DMF (or dimethyl sulfoxide-DMSO) and $CH_3CN$ system. These alloys presented potential for application as magnetic, photo-electronic, thermoelectric and fuel cell materials".

Our group "investigated AlCrFeMnNi and AlCrCuFeMnNi high-entropy alloy thin films prepared by potentiostatic electrodeposition in an electrolyte based on a DMF (N,N-dimethylformamide)-$CH_3CN$ (acetonitrile) organic compound" [25]. For corrosion protection, these new HEA alloys have a promising potential [26].

Recently, other studies have focused on the pulse electrodeposition method to obtain HEA for various applications [27,28].

The present study demonstrates the possibility of electrodeposition synthesis of thin films of alloys with high-entropy CoCrFeMnNi. It also shows the behavior of these alloys to corrosion compared with the copper support. As no reports of a similar study have been made so far, we were not able to compare the obtained results and consider them to be new.

## 2. Materials and Methods

HEA electrodeposition and corrosion studies were performed at 298 K. Electroplating was performed on a copper support (2 $cm^2$ thin plates) with a Princeton Applied Research 263A potentiostat/galvanostat (AMETEK Princeton Applied Research, Oak Ridge, TN, USA). The electrolyte used for potentiodynamic electrodeposition consisted of an organic medium consisting of 4:1 volume fraction of DMSO-$CH_3CN$, to which were added $LiClO_4$, $Co(NO_3)_2$, $CrCl_3$, $FeCl_2$, $MnCl_2$, and $NiCl_2$. Table 1 presents the composition of the used electrolyte.

**Table 1.** Chemical composition of the electrolyte.

| $Co(NO_3)_2$ mol/L | $CrCl_3$ mol/L | $FeCl_2$ mol/L | $MnCl_2$ mol/L | $NiCl_2$ mol/L |
|---|---|---|---|---|
| 0.014 | 0.016 | 0.012 | 0.012 | 0.012 |

The electrochemical cell used was a classic one with three electrodes: the counter electrode was a Pt plate (2 $cm^2$), the reference electrode was a calomel electrode saturated with KCl (SCE, +0.241 V), and the working electrode was Cu for electrodeposition and Cu with HEA coatings for corrosion measurements. Prior to electrodeposition, the Cu substrates were thoroughly prepared by polishing with abrasive paper of various sizes, washing with a 67% $HNO_3$ solution, and finally rinsing with double-distilled water.

SEM images and EDS spectra were obtained by using Quanta 3D FEG D9399 equipment (FEI Company, Hillsboro, OR, USA) (gallium ion source). Those experiments were used to analyze surfaces and to obtain the chemical composition of the HEA thin films.

Atomic force microscopy (AFM) measurements were performed in non-contact mode with XE-100 from Park Systems (Park Systems Corp., Suwon, Korea), equipped with flexure-guided, crosstalk eliminated scanners. For sharp tips, PPP-NCLR from Nanosensors™ (Nanosensors™, Lady's Island, SC, USA), having less than 10 nm radius of curvature, ~225 mm length, ~38 mm width, ~48 N/m force constant, and ~190 kHz resonance frequency was used to record the AFM images. XEI program (v 1.8.0-Park Systems) was used

for displaying purpose and roughness evaluation of images. The "enhanced contrast" view mode was used to improve the topographic details. AFM 2D images show representative line scans. These represent in detail the surface profile of the scanned samples.

The corrosion tests were performed on Voltalab 80 PGZ 402 equipment (Radiometer Analytical SAS, Lyon, France) with a special Corr. soft. In order to prevent electrical interferences, the cell assembly was placed in a Faraday cage.

## 3. Results and Discussion

Figure 1 shows the EDS analysis and Table 1 shows the chemical compositions of the HEA alloys studied at different deposition times. These results confirm that all the studied deposits contain all five elements.

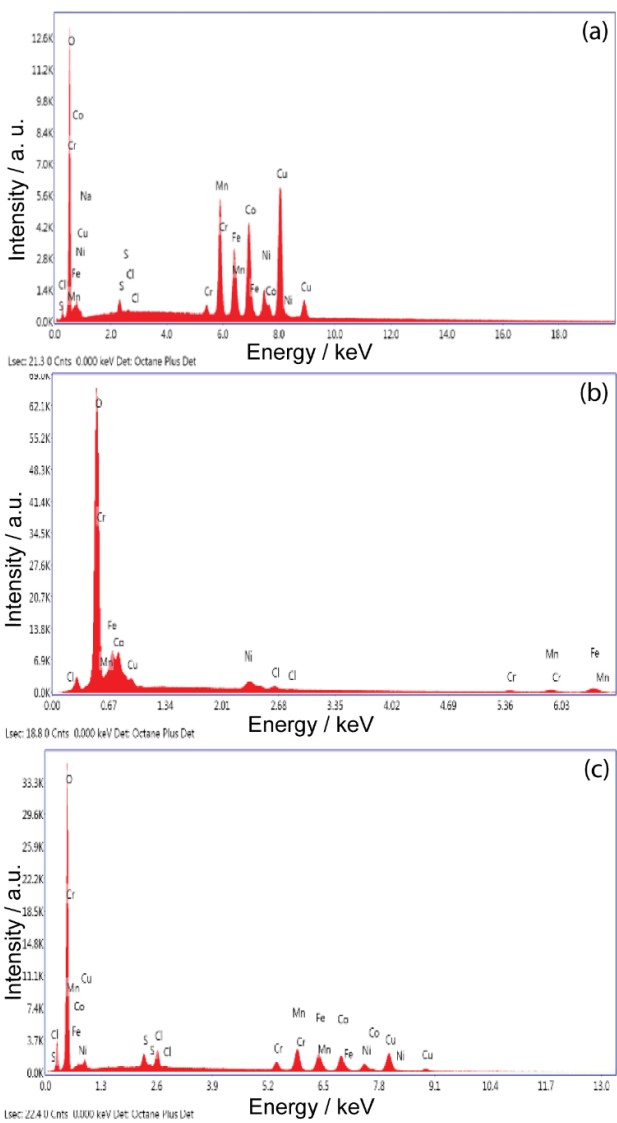

**Figure 1.** EDS spectra for the CoCrFeMnNi alloy deposited at −2.5 V for different periods of time: (**a**) 30 min; (**b**) 60 min; (**c**) 90 min.

The depth profile of the EDS spectrum suggests that the chemical composition of the deposit is relatively constant from the bottom to the top of the sample.

The maximum value of the atomic percentage of Co (40.13%) from the studied alloys is found for HEA obtained at a deposition time of 60 min and at −2.1 V. The increase in the deposition time is caused by the reduction in the Co and Fe content, followed at the same time by the increase in the content of the other elements from the HEA alloy.

Electrolyte composition and processing parameters during electrodeposition are determinants of alloy composition However, in this case, the chemical composition is determined by the anomalous deposition behavior of Mn, which is among the elements of this HEA system. Standard reduction potentials of the five metals in the present HEA system follow the order Ni > Co > Fe > Cr > Mn. However, preferential deposition of higher amounts of Mn is observed, rather than Ni. The easy formation of metal hydrate ions, hydroxide/ligands (ligands form as a result of complexing agents), and its quick dissociation in the reduction process, in addition to multiple reduction sequences at the cathode surface, can be the explanation for this behavior.

Figures 2–4 present the SEM images of the CoCrFeMnNi alloy thin films deposited at −2.5 and −2.1 V for 30, 60, and 90 min. The HEA thin films deposited consist of particles with sizes ranging from 5 µm to ~50 nm (Figures 2b and 4b). The shapes of these particles are spherical, grain shaped, and conglomerates. The HEA thin films obtained at an electrodeposition potential of −2.5 V (Figure 2) are homogenous and present visible microcracks. Less visible cracks but with uniform and homogenous distribution of spherical particles are observed for the HEA thin films obtained at −2.1 V for 60 min. Increasing the deposition time to 90 min leads to the formation of grain-shaped particles with lower mean size. It may be said that a longer processing duration determines a refinement of thin-film morphology.

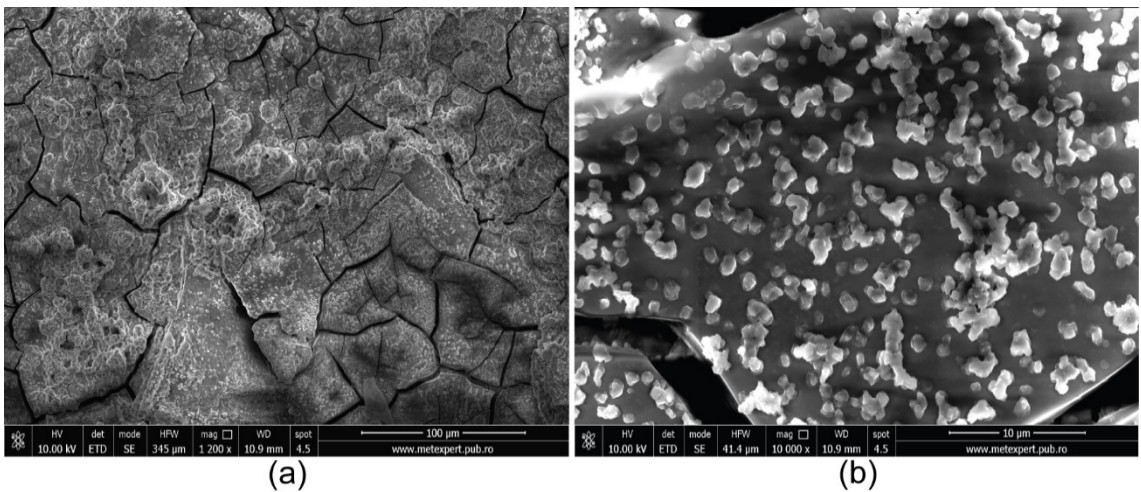

**Figure 2.** SEM morphology of the CoCrFeMnNi film electrodeposited at −2.5 V for 30 min: (**a**) ×1200; (**b**) ×10,000.

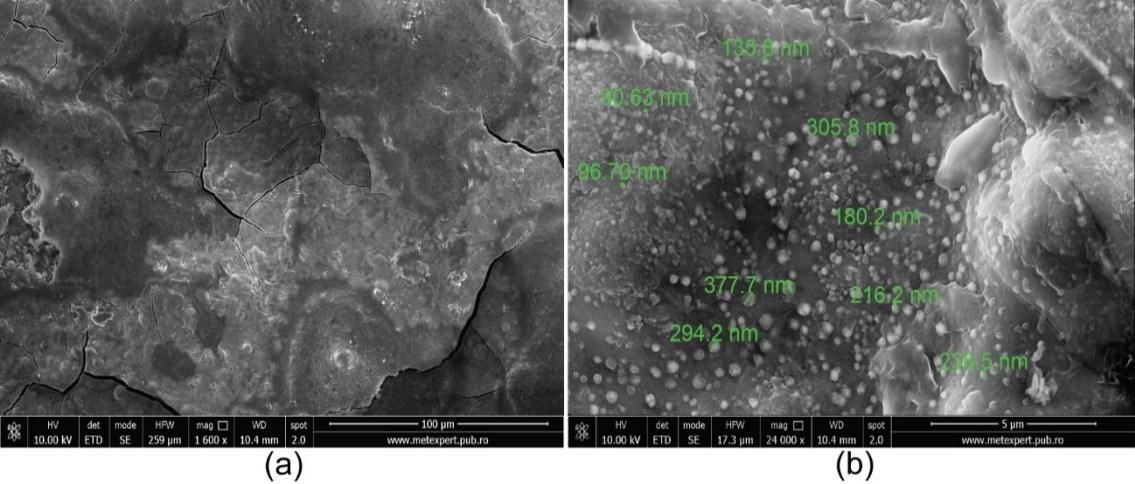

**Figure 3.** SEM morphology of the CoCrFeMnNi film electrodeposited at −2.1 V for 60 min: (**a**) ×1600; (**b**) ×24,000.

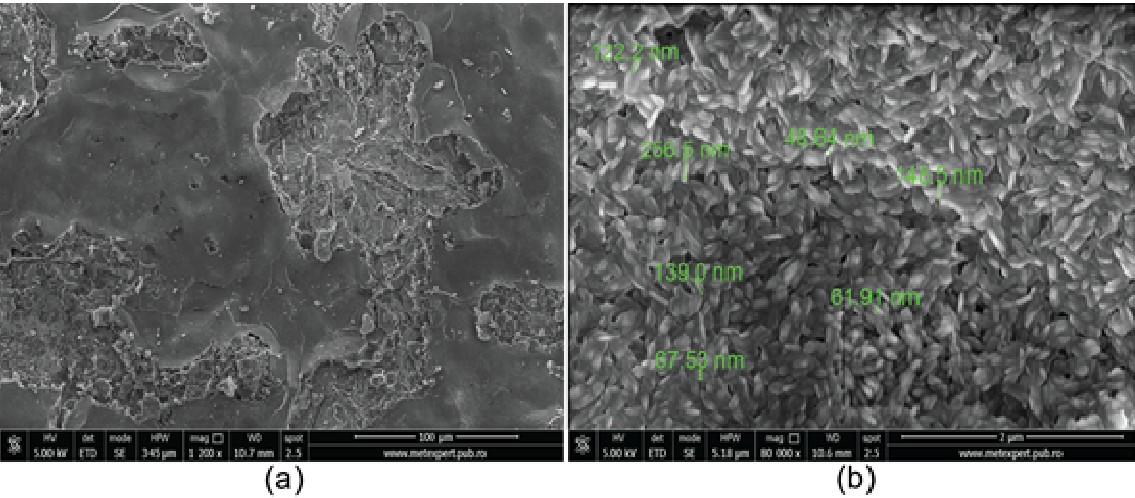

**Figure 4.** SEM morphology of the CoCrFeMnNi film electrodeposited at −2.1 V for 90 min: (**a**) ×1200; (**b**) ×80,000.

From these SEMs, one can observe that the number of the wheat grain formations increases with the decrease in the deposition potential and increase in the process duration. The particles that form the high-entropy alloy film are functions of the variation in the electrodeposition potential and deposition time. The optimal value for the electrodeposition potential is −2.1 V. This is concluded from the SEM images and chemical compositions in Table 2.

**Table 2.** HEA thin films chemical composition for electrodeposition at different potentials for 30, 60, and 90 min.

| Deposition Time min | Potential V | Chemical Percentage at. % | | | | |
|---|---|---|---|---|---|---|
| | | Co | Cr | Fe | Mn | Ni |
| 60 | −2.1 | 40.1 | 6.8 | 32.9 | 16.8 | 3.4 |
| 90 | | 26.2 | 8.6 | 23.7 | 27.2 | 14.3 |
| 30 | −2.5 | 33.2 | 2.7 | 19.7 | 32.7 | 11.7 |

Electrochemical measurements for the uncoated/coated copper samples were performed in an aerated artificial seawater solution under ambient conditions.

Polarization curves of uncoated copper and HEA-coated copper are presented in Figure 5. It can be observed from Figure 5 that both cathodic and anodic polarization curves denote lower current density in the presence of HEA coating. This behavior reveals that the HEA coating has a significant effect on the cathodic and anodic reactions of the electrochemical process. The Tafel plot is presented in Figure 5, while Table 3 presents the measured open-circuit potential (OCP), corrosion potential ($E_{corr}$), and corrosion current density ($i_{corr}$), as well as the calculated corrosion parameters—polarization resistance ($R_p$), corrosion rate (CR), and penetration index (PI). The results show a good corrosion resistance for the HEA thin film in artificial seawater. It is obvious that the HEA coating impedes the attack of the aggressive ions ($Cl^-$) on the electrode surface.

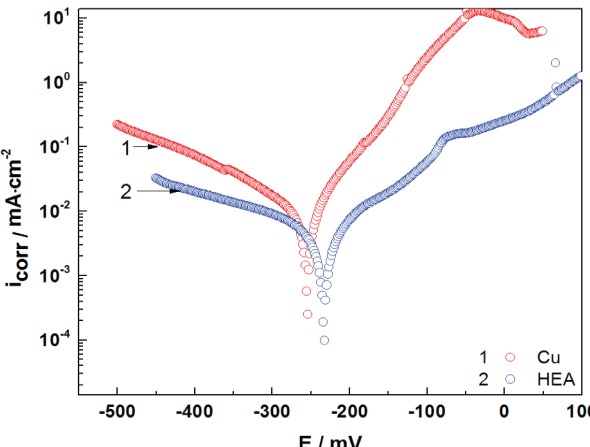

**Figure 5.** Polarization curves of uncoated Cu and HEA-coated copper in artificial seawater at 25 °C.

**Table 3.** The corrosion parameters of the Cu and Cu/HEA in artificial seawater at 25 °C.

| Sample | $E_{OCP}$ mV | $R_p$ $\Omega$ cm$^2$ | $E_{corr}$ V | $i_{corr}$ $\mu$Acm$^{-2}$ | CR mpy | PI mm y$^{-1}$ |
|---|---|---|---|---|---|---|
| Cu | −212 | 1780 | −260 | 7.814 | 3.65 | 0.092 |
| Cu/HEA | −180 | 5040 | −236 | 3.303 | 2.011 | 0.051 |

$E_{OCP}$ = potential at zero current; $R_p$ = polarization resistance; $E_{corr}$ = corrosion potential; $i_{corr}$ = corrosion current density; CR = corrosion rate; PI = penetration index.

The corrosion behavior, corrosion mechanisms, and adsorption phenomena of the films formed on the surface of the CoCrFeMnNi alloy coating sample in artificial seawater solution were studied by electrochemical impedance spectroscopy (EIS). The EIS measurements were carried out over a frequency range of 100 kHz–40 mHz at open-circuit potential (OCP) after 60 min immersion in seawater, with a sinusoidal AC voltage waveform of ±10 mV (peak-to-peak). The results show the electrochemical properties of the copper/alloy/electrolyte interface.

The Nyquist diagrams for uncoated Cu and HEA-coated copper electrodes in artificial seawater are presented in Figure 6. EIS data were examined utilizing equivalent electrical circuits, as shown in Figure 7.

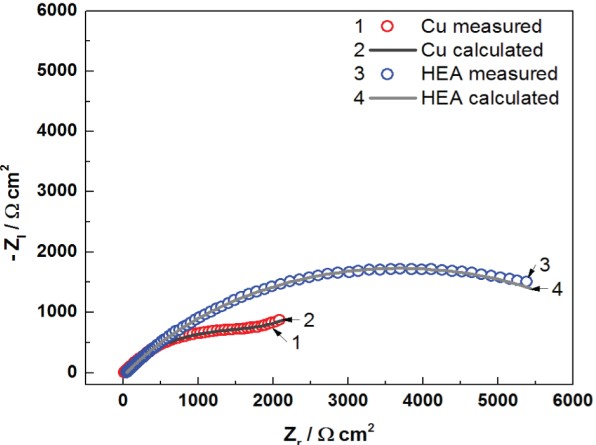

**Figure 6.** The Nyquist diagrams of the Cu and Cu/HEA electrodes in artificial seawater. Lines represent the modeled data according to the EECs from Figure 7.

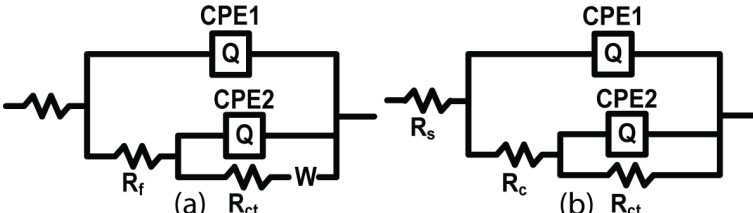

**Figure 7.** The equivalent circuit model used to fit the experimental data (**a**) for uncoated Cu and (**b**) for Cu/HEA coating in artificial seawater.

It can be seen from Figure 6 that the impedance plot of uncoated Cu can be depicted by two time constants. The first time constant at a high-frequency interval is associated with the presence of corrosion film established on the copper surface, where $R_f$ is the resistance of the corrosion film, and CPE1 is the capacitance of the corrosion film. The second time constant, at middle and low frequencies, constitutes charge transfer resistance ($R_{ct}$) and capacitance of the double layer (CPE2) in series with Warburg impedance (W). The element Warburg indicates a diffusion process throughout immersion of the Cu electrode in the artificial seawater. It is difficult to evaluate the EIS data for Cu electrodes, as results are subject to various interpretations. However, the linear aspect of the Nyquist plot suggests a semi-infinite diffusion type for the Cu electrode [29]. The Warburg impedance considers the diffusion of oxygen and diffusion of corrosive species (as $Cl^-$) to the copper surface and/or diffusion process of soluble copper compounds ($CuCl_2^-$) from the surface into the solution [29–33]. The appearance of diffusion is also considered in the Bode diagram (phase angle vs. log f), where the phase angle amounts around $(-45°)/(-50°)$. Therefore, the EIS response for the Cu–seawater interface is depicted by the electrical equivalent circuit (EEC) with two time constants (Figure 7a).

The Nyquist plot of HEA coating is shaped by utilizing approximately the same equivalent circuit exhibited in Figure 7. In addition, the parameter values with very good fit to the impedance plots are meaningfully distinct in comparison to those acquired for uncoated copper. In this occurrence, the first capacitive loop was assigned to the features HEA coating/electrolyte interface, and it is distinguished by the coating resistance ($R_c$) and coating capacitance CPE1. The second one in the middle- and low-frequency interval was ascribed to the HEA–Cu interface and the processes occurring underneath the coating; it is characterized by the charge transfer resistance ($R_{ct}$) and double-layer capacitance (CPE2). It is clear from the Nyquist plots that the impedance response of copper is significantly modified by the deposition of the HEA coating, which means a protective film is achieved, as evinced by the presence of HEA.

Evidently, a very good fit with this model is realized as regards all experimental impedance data, which is in good agreement with data obtained from the potentiodynamic polarization measurements.

Furthermore, the greater values of $R_{ct}$ and lesser values of $C_{dl}$ denote the best corrosion protection performance of the HEA-coated copper electrode in artificial seawater. The parameters achieved by fitting the equivalent circuit of the uncoated Cu and HEA coating are presented in Table 4. Therefore, in the studied frequency range, an equivalent circuit model (Figure 7) was proposed, resulting from the fitting and analysis of EIS experimental data. In this case, the phase element constant CPE was introduced into the circuit instead of a pure double-layer capacitor ($C_{dl}$) to provide a more accurate fit.

$$C_{dl} = Y_0 \left(\omega_{max}\right)^{n-1} \tag{1}$$

**Table 4.** The EIS parameters of the Cu and Cu/HEA in artificial sea water.

| Sample | $R_s$ ohm·cm² | CPE1 | | R * ohm·cm² | CPE2 | | $R_{ct}$ ohm·cm² | W S·s$^{-1/2}$·cm$^{-2}$ | $\chi^2$ | $R_p$ ohm·cm² | E% |
|---|---|---|---|---|---|---|---|---|---|---|---|
| | | Q-Yo S·s$^{-n}$·cm$^{-2}$ | Q-n | | Q-Yo S·s$^{-n}$·cm$^{-2}$ | Q-n | | | | | |
| Cu | 16.95 | $1.143 \times 10^{-4}$ | 0.705 | 232 | $1.642 \times 10^{-4}$ | 0.622 | 1843 | 0.002246 | $6.176 \times 10^{-4}$ | 2075 | - |
| Cu/HEA | 41.54 | $7.485 \times 10^{-6}$ | 0.755 | 48.2 | $7.717 \times 10^{-5}$ | 0.635 | 7339 | - | $6.852 \times 10^{-5}$ | 7387 | 72 |

\* R for Cu is $R_f$—the corrosion film resistance; \* R for Cu/HEA is $R_c$—the coating resistance.

CPE was used to describe the deformation of the capacitive semicircle, which corresponds to the heterogeneity of the surface due to roughness and impurities.

The CPE impedance can be defined as:

$$Z_{CPE} = Y_0^{-1} (j\omega)^{-n} \tag{2}$$

where $\omega$ is the angular frequency ($\omega = 2\pi f$), j is the imaginary number ($j^2 = -1$), $Y_0$ is the amplitude comparable to capacitance, and n is a phase change. The value of *n* provides details about the degree of inhomogeneity of the metal surface. A higher value of *n* is associated with a lower surface roughness (a reduced inhomogeneity).

The phase element constant can be set as a resistance (when n = 0, $Y_0$ = R), a capacitance when (n = 1, $Y_0$ = C), an inductance (when n = −1, $Y_0$ = 1/L), or a Warburg impedance (n= 0.5, $Y_0$ = W), depending on the value of n.

An increased value of the charge transfer resistance ($R_{ct}$) may be attributed to the formation of the protective film at the metal–solution interface and a small value for the double-layer capacity ($C_{dl}$) may be due to the decrease in the local dielectric constant or the increase in the electric double-layer thickness, indicating a protective behavior of the HEA surface. The values of polarization resistance and related percentage protection efficacy values were determined from EIS data and are presented in Table 4. The anti-corrosion protection efficiency was calculated from polarization resistance using the following relation:

$$\%\text{Efficiency} = (R_{pc} - R_p/R_{pc}) \times 100 \tag{3}$$

where polarization resistance $R_p$ from EIS measurements can be calculated as $R_p = R_{ct} + R_f$ (polarization resistance for copper) and $R_{pc} = R_{ct} + R_c$ (polarization resistance for HEA-modified copper). The percentage efficiency calculated from EIS data is found to be 72%. This is in agreement with the potentiodynamic polarization results.

It can be inferred that the higher values of $R_p$ for HEA coating are attributed to the efficacious protection of HEA, and the lower value of CPE for HEA coating assures support for the anti-corrosion protection of copper by coating HEA in artificial seawater.

It can be observed that the diameter of the electrochemical impedance loop for Cu/HEA is greater than that for uncoated copper, which points to better anti-corrosion properties of the HEA-modified copper electrode, compared with the unprotected copper.

The Bode diagrams shown in Figure 8 are consistent with the Nyquist plot, and it can be observed that a higher value of the impedance modulus ($Z_{mod}$) at low-frequency values describes a better protective efficiency of the surface against corrosion.

From Figure 8 (Bode diagrams), it can be observed that the presence of HEA–Cu on the graph-phase angle against the frequency logarithm, presents a maximum very well established at a phase angle of approximate −50° for HEA; hence, this event indicates the presence of microdefects (microcracks) in the coating structure. The Nyquist and Bode diagrams suggest that the film on the surface of HEA hinders the corrosion process and acts as an impediment for the charge transfer phenomenon. Additionally, the EIS data are in good agreement with the potentiodynamic polarization results. In this study, we analyzed the behavior of the obtained HEA thin film at corrosion state, as described above. Results from this study show a good protection efficiency of HEA coating deposited on the surface of copper electrodes in seawater. Research for improving anti-corrosive protective

properties of thin-film HEA will continue, seeking to investigate and observe the results over a longer period of time.

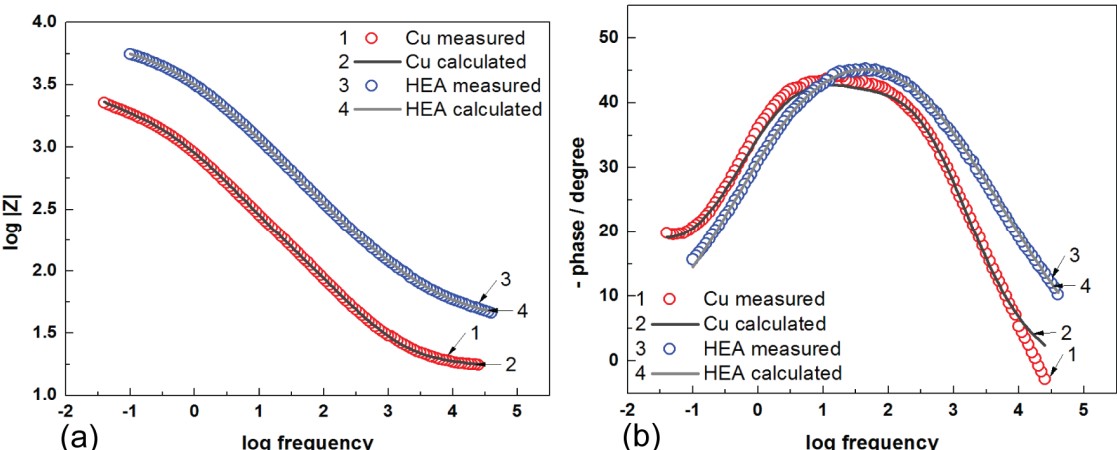

**Figure 8.** The Bode diagrams: (**a**) log freq. vs. log|Z| and (**b**) log freq. vs.–Phase, of the Cu and Cu–HEA electrodes in artificial seawater. Lines represent the modeled data according to the EECs from Figure 7.

The protective behavior of the HEA coating is illustrated by the AFM images.

In order to obtain information about the morphology of the samples and their roughness, AFM investigations were performed on the CoCrFeMnNi coating sample before (1) and after corrosion (2) in artificial seawater. Figure 9 comparatively presents 2D-AFM and 3D-AFM images at the scale of (4 μm × 8 μm) for Sample 1 (HEA initial) and Sample 2 (HEA corroded), together with random plotted line scans.

As can be seen from Figure 9a, Sample 1 (HEA initial) exhibits hills-like surface features mainly located in a vertical domain from −100 to +100 nm, as can be observed from the corresponding line scan presented below the 2D AFM image, and random pits (see the yellow arrows), which are about 200 nm deep. The scanned area (4 μm × 8 μm) is 57.9 nm (59.4 nm RMS roughness along the red line), while the peak-to-valley parameter ($R_{pv}$) is 483.5 nm (317.0 nm along the red line).

Due to the corrosion process, Sample 2 (HEA corroded) exhibits a curved profile, with surface features mainly located in z axis from −0.25 to +0.25 μm, as visible in the line scan presented below the AFM image (it can be assumed that due to the corrosion process, the corrugated profile visible in Figure 9b is filled with the corroded materials, which leads to a less spiky but curved surface profile). The global RMS roughness of the scanned area is 306.6 nm (287.6 nm RMS roughness along the red line) for Sample 2, while the peak-to-valley parameter ($R_{pv}$) is 1.74 μm (1.40 μm along the red line). However, both 2D and 3D AFM images of Sample 2 suggest that the corrosion process is continuous, with a thickness in the micrometer range.

A better view of the morphology was obtained at the scale of (3 μm × 3 μm), as presented in Figure 10. The surface of Sample 1 is corrugated, but the presence of small grains of about 50 nm in diameter is observed, as indicated by the particle selected between the two red arrows. The characteristic line scan plotted below the 2D AFM image, indicates spiky hills and some valleys in between, located in a vertical height of ~160 nm (from −80 to 80 nm). The RMS roughness of Sample 1 is 32.7 nm, while the peak-to-valley parameter is 273.8 nm.

Sample 2 suggests the formation of corrosion materials during the corrosion process, so that globular large clusters of >150 nm are visible on the surface (see the selected globular-shaped deposit with 164 nm diameter). The RMS roughness of sample 2 is 44.54, and the peak-to-valley parameter is 325.7, which are larger than the values of the "parental" surface (Sample 1), due to the corrosion products formed by the corrosion process.

Finally, on the same sample (presented in the initial state in Figure 4) subjected to corrosion in synthetic seawater and AFM measurements, SEM micrography was performed

(Figure 11). SEM images of the HEA-corroded sample proved by AFM measurements that the corrosion process does not considerably change the morphology (Figure 11a). At higher magnifications (Figure 11b), the initial grain-shaped particles modifies to globular ones, which means that the surface has more roughness.

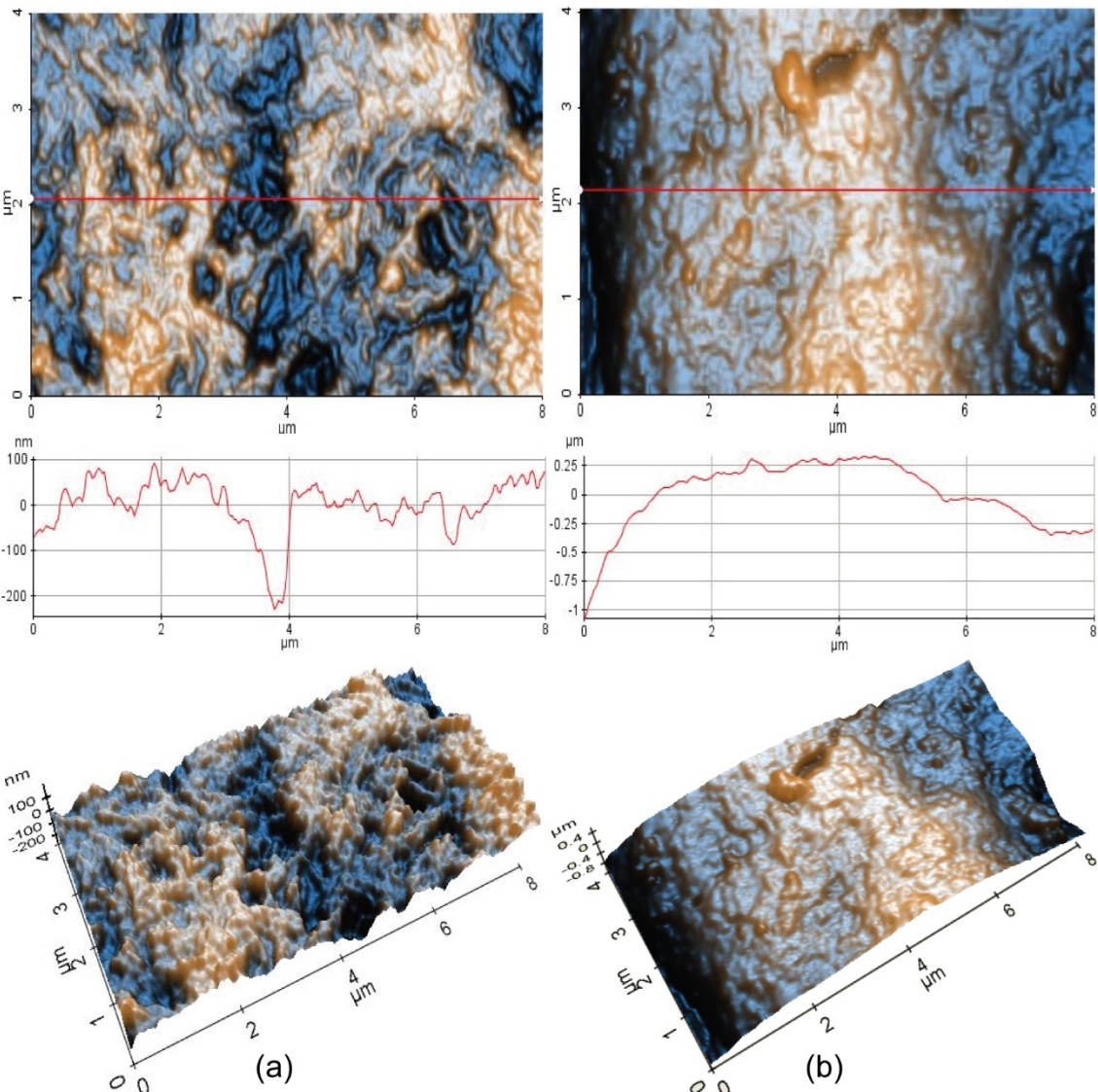

**Figure 9.** The 2D and 3D AFM topographic images of Sample 1 (**a**) and Sample 2 (**b**) at the scan area of (4 μm × 8 μm); representative line scans are shown, collected at the position indicated by a horizontal red-line in the 2D AFM images.

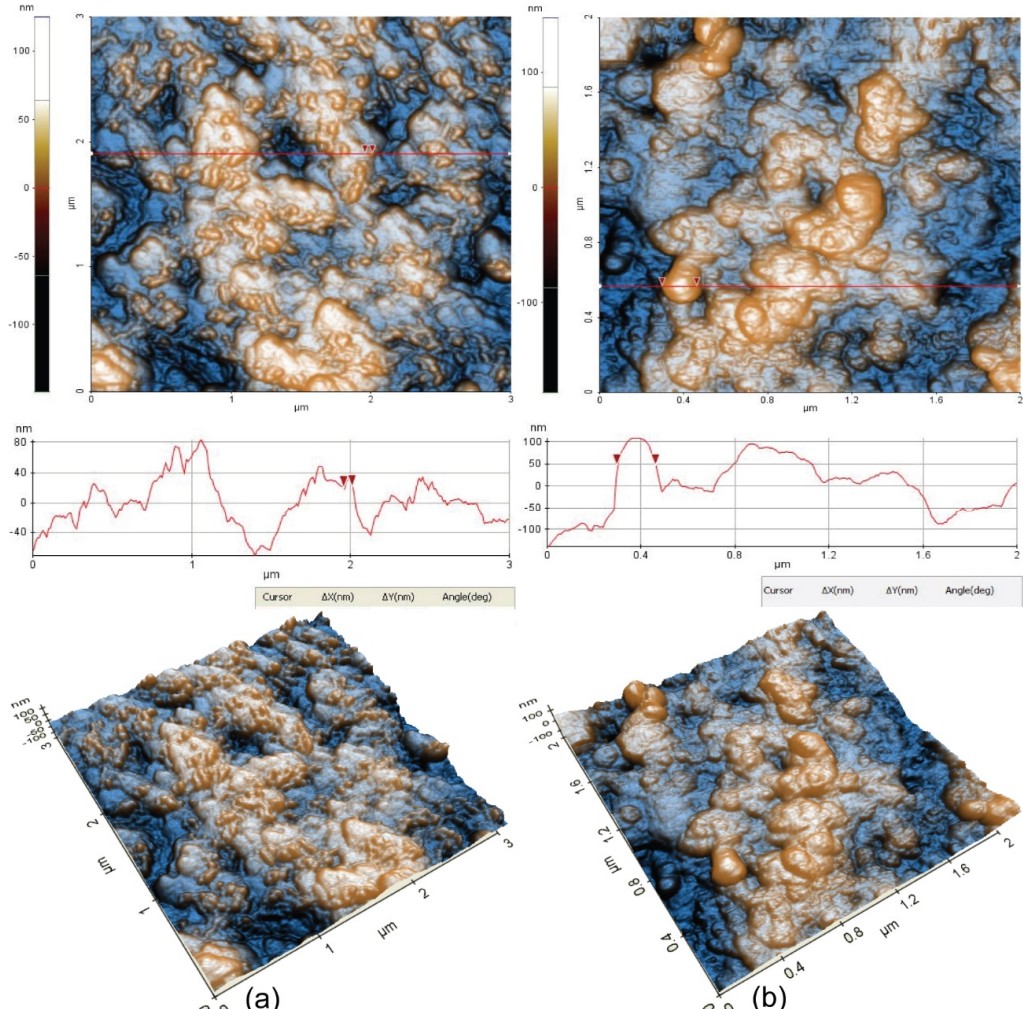

**Figure 10.** The 2D and 3D AFM topographic images of Sample 1(initial) (**a**) and Sample 2 (corroded) (**b**) at the scale of (3 μm × 3 μm); representative line scans are shown, collected at the position indicated by a horizontal red-line in the 2D AFM images.

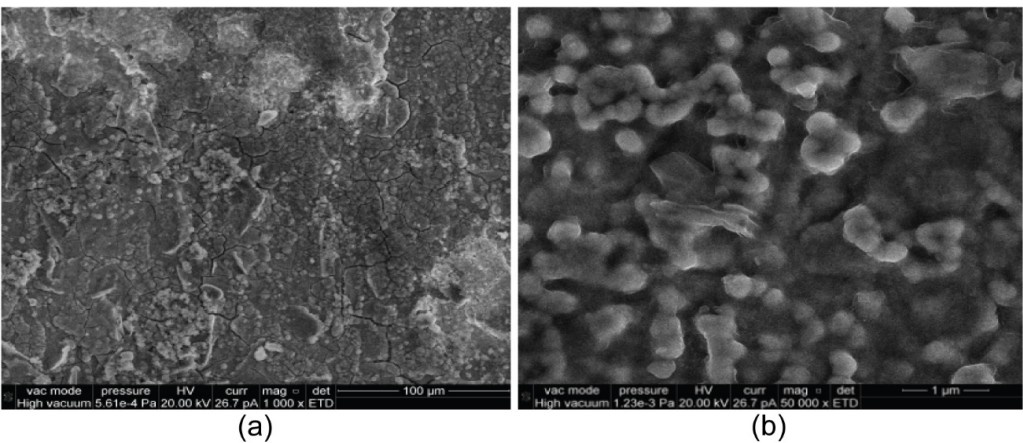

**Figure 11.** SEM morphology of the CoCrFeMnNi (film electrodeposited at −2.1 V for 90 min) corroded in artificial seawater: (**a**) ×1000; (**b**) ×50,000.

## 4. Conclusions

CoCrFeMnNi high-entropy alloys were prepared by potentiostatic electrodeposition at various potentials in the DMF-CH$_3$CN organic system with LiClO$_4$ additive.

The presence of all elements in the HEA coating is highlighted by the EDS results. SEM surface analysis reveals that HEA thin films consist of three types of particles (spherical, grain shaped, and clusters), with sizes ranging from ~50 nm to 5 μm. It is also revealed that the HEA thin films deposited at a potential of −2.1 V exhibit better growth and more homogeneous distribution of the particles compared with the samples deposited at −2.5 V.

Combining SEM and EDS results, we can conclude that by the variation in the deposition potential and time, we can control the surface morphology and chemical composition of HEA alloy.

The corrosion and AFM studies prove that the deposited CoCrFeMnNi high-entropy alloy electrode has a good corrosion behavior in artificial seawater.

These studies demonstrate that the electrodeposition in organic media can be used for the synthesis of HEA multi-element alloys with good surface morphology and corrosion behavior.

In this study, we analyzed the behavior of the obtained HEA thin film at corrosion state, as described above. The HEA film provides additional protection to the copper electrode during immersion in artificial seawater (corrosion efficiency of 72%), but imperfections present in the film structure could influence the corrosion behavior during a long period of immersion in this aggressive electrolyte. These imperfections might become wider with more aggressive ions/molecules and could penetrate inside the film, and consequently, the HEA anti-corrosion protection would change, but this conclusion could be demonstrated only by long-term corrosion studies. Consequently, in order to improve the deposition method and corrosive protection properties of HEA films, future studies are planned.

**Author Contributions:** Conceptualization, A.-M.J.P. and I.C.; methodology, F.B.; software, M.-T.O.; validation, M.B., I.A. and D.M.; formal analysis, F.B. and V.C.; investigation, F.B., M.A. and M.B.; resources, A.-M.J.P., writing—original draft preparation, A.-M.J.P. and I.C.; writing—review and editing, A.-M.J.P.; visualization, F.B.; supervision, V.C.; project administration, A.-M.J.P. and M.B.; funding acquisition, A.-M.J.P. and M.B. All authors have read and agreed to the published version of the manuscript.

**Funding:** This research was funded by Romanian Executive Agency for Higher Education, Research, Development and Innovation Founding (UEFISCDI), project HEASYNTCORR/PN-III-P2-2.1-PED-2019-0022, contract 330PED/2020.

**Institutional Review Board Statement:** Not applicable.

**Informed Consent Statement:** Not applicable.

**Data Availability Statement:** Not applicable.

**Conflicts of Interest:** The authors declare no conflict of interest.

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
