# Peer review of "Electrodeposition, Characterization, and Corrosion Behavior of CoCrFeMnNi High-Entropy Alloy Thin Films"

_coatings, doi:10.3390/coatings11111367_

Round 1

Reviewer 1 Report

see attached file

Author Response

I made all correction.

Reviewer 2 Report

The paper is focused on preparation and characterization of CoCrFeMnNi high entropy alloys (HEA) as promising anti-corrosion protective films. In the manuscript there are some mistakes and missing information, therefore a major revision is needed.

Remarks:

Materials and Methods:

-lines 84-85: -"standard?? saturated calomel electrode

-It is evident from the results that the film prepared at the potential of -2.5 V was chosen asd "the best formed film".  Why ?
It would be useful to see the responses of electrodeposition recorded at all investigated potentials. Please comment.

Results and discussion:

-Fig. 1 caption: X- and Y- axes?? What they present?

-Fig. 1 shows a strong signal of oxygen and copper which probably indicates a presence of copper oxide on the copper surface. Signal of Cl is also visible. If this is a spectrum of freshly prepared alloy on the copper surface, it is very strange that the Cl is present. EDS spectrum reveals a signal of S. Please comment.

-Figs. 2-4: Spherical particles - their chemical composition and flat parts of the surface - chemical composition? EDS analysis should reveal their chemical composition. Obviously the film is not defect-free. 

-Lines 152-168: Polarisation curves: Fig. 5 caption: incomplete and unclear. This fig. shows the comparison of the polarisation  curves of unmodified and modified copper electrodes.

-Line 152: steel samples?

-To get deeper insight in the corrosion behaviour of all prepared films, it would be useful to show the Tafel plots of all prepared samples and then evaluate their anti-corrosion properties in an electrolyte solution.

-Line 171: EIS = electrochemical impedance spectroscopy

-Lines 179-180: ...."with the charge transfer process of the film on the electrode surface." What does it mean? Charge transfer process occurs at the copper/alloy/electrolyte interface and charge transfer can occur through the film.

-The whole section from line 186-202 is unneccessary. Since surface inhomogeneity (lines 183-185) influence on impedance responses, capacitance have to be replaced with CPE. Therefore the data in Table 4 are incorrect for the evaluation of corrosion parameters. 

As can be seen from Fig. 6 (Niquist diagram) the EIS response of both electrodes are more complex and can not be described by the EEC with one-time constant. This is especially reflected in the behaviour of the copper electrode, which has, at low frequencies, additional time-constant that is ommited in the phase angle vs. frequency dependence (Fig. 8). It is well-known from numerous published papers that the copper electrode in the electrolyte used has complex EIS response (2 or more time constants).

Fig. 8 , phase angle vs. frequency dependence indicates a presence of diffusion process, since phase angle values, at middle frequencies, are around -45. The phase angle is a structural-sensitive parameter and reflects a presence of defects/pores in the investigates systems. As can be seen (Fig. 8) electrode covered with alloy has also a low values of phase angles which points to film with structural defects. As stated in the text, parameter n of the CPE will also indicate a present of diffusion proces.
Lines 219-224: "The Bode diagrams shown in Fig. 8 are consistent with the Nyquist plot and it can be observed that a distinctive peak is present on the phase angle vs frequency logarithm plot at a phase angle value of approximately 50º, which represents a remarkable capacitive behavior...." are completely wrong. Only phase angle values around 80-90º indicate a real capacitive bahaviour. 

Therefore, EIS part should be completely modified and rewritten. The new EEC should be used and corrosion parameters should be calculated from the fiiting parameters and then should be correlate with polarization results. Charge transfer resistance will be given by fitting procedure. For the investigated system and evaluation of corrosion parameters a key parameter is the polarization resistance, Rp.

-AFM results also indicate a presence of pores/defects in the prepared ally when surface roughness of bare copper is compared with roughness of alloy-modified surface.

Author Response

I made all correction

Round 2

Reviewer 2 Report

The authors did not answer the previous questions.

In the new part of the EIS, although new EECs have been selected, there are fundamental errors. It is necessary to improve the text and correlate with the polarization results because anti-corrosion characteristics were determined with both methods.

It should be taken into account that the real corrosion behavior in an agressive environment like seewater should be investigated over a period of time and the results obtained should be put in that context.

Define more precisely what is the capacity of the double layer, charge transfer resistance, and what is the capacity of the film (in the case of alloy) or the capacity of the corrosion product (in the case of copper) in selected EEC circuits. What is the polarization rate (line 156; Tafel results)? Please explain.

Fig. 8. is not Bode diagram. Bode diagram represents log Z vs. log f and phase angle vs. log f. 

It would be useful to see fitting curves in Figs. 7 and 8.

The following text: "The Nyquist and Bode diagrams suggest that the film on the surface of the alloy halted the corrosion process and acted as a diffusion barrier created through the charge transfer phenomenon" is not completely correct (a part of the sentence related to the diffusion barrier). Data in Table 4, especially a fine-structural parameter n of the CPE, which is low (around 0.6 or 0.75), indicates structural defects inside the alloy formed. These defects will be crucial for alloys corrosion behaviour in very aggressive environment like artificial seewater (high amount of cloride ions).

Author Response

I rewrite and corrected MS, in accordance with the recommendations of Prof. Rew 2.

Round 3

Reviewer 2 Report

The points for further improving of the manuscript:

References related to copper corrosion properties in artificial see water are missing.

Table 2: The sum of all at.% at every potential should be 100%.

Lines 154-155 should be improved: ....while Table 3 presents the measured [open circuit potential (OCP), corrosion potential (Ecorr); and corrosion current density (icorr)] and calculated corrosion parameters [polarization resistance (Rp), corrosion rate (CR), and penetration index (PI)].

Table 3: a) Please correct symbol for corrosion rate; instead of R should be CR; b) Unit of Rp (Wcm-2) is wrong. It should be (W cm2).

Lines 171-173: Stabilization period before EIS measurements should be added. „The EIS measurements ....at open circuit potential after (min/hours??) stabilization with an ac voltage....

Lines 173-174: The sentence is not completely correct. Since EIS measurements are performed in the wide frequency range, the results obtained reflect all interfaces present in the system (copper/alloy and alloy/ electrolyte). Therefore, it would be more correct to write: : The results show the electrochemical properties of the copper/alloy/electrolyte interface.

General remark: Two electrodes investigated, bare or uncoated copper electrode and the HEA-coated copper electrode, should have uniform names through whole manuscript.

Figures 6 and 8: a) Figure 6 and 8 represent the same results and it would be clearlier to present them together as one figure and comment togheter in the text.

b)Figs. captions: Please correct: The Nyquist diagrams of the Cu and Cu/HEA electrodes in ..... Lines represent the modelled data according to the EECs from Fig. 7.  c) Text in Table in Figs. should contain Cu and Cu/HEA as „names“ for the investigated electrodes.

Figure 7: Please correct: .....a) for uncoated Cu and b) for Cu/HEA  in ....

Lines 177-189 : The text about EIS response of bare Cu is unclear and a part connected with diffusion and Warburg element is partialy correct. REFERENCES SHOULD BE CITED, SINCE THIS Cu BEHAVIOUR IS WELL REPORTED IN THE LITERATURE.

Explanation: Bare Cu response can be described by two time constant. The first at high frequencies is connected to the presence of corrosion film formed on the copper surface, where Rf is the resistance of the corrosion film and CPE1 is capacitance of the corrosion film. The second time constant, at midlle and low frequencies, represents charge transfer resistance (Rct) and capacitance of the double layer (CPE2) in series with Wargurg impedance. The element Warburg indicates a diffusion process during immersion of the Cu electrode in the artificial see water. It can be connected with oxygen diffusion oand corrosive species diffusion (like Cl-) to the Cu surface and/or diffusion of copper species from the cu surface (additional measurements are mandatory in order to determine which diffusion is present!!!). The appearence of diffusion is also reflected in the Bode diagram (phase angle vs. log f) where phase angle amounts around -45 – (-50)°. Therefore the EIS response for the Cu/see water interface is described by the electrical equivalent circuit (EEC) with two time constant (Fig. 7a).

Lines from 207-218 about EECs should be connected with the text 177-202.

Before Table 4 with modelled parameters should be the text (lines 234 -254) why capacitance (C) is presented as CPE (Q) parameter.

Lines 194-195: ....The symbols used in EEC for the HEA-copper  should be define in the text; the HEA coating resistance (Rc) and coating capacitance (CPE1).

Lines 196-197: charge transfer resistance (Rct) and double layer capacitance (CPE2)

Lines 201-202: “It can be noticed that, the diameter of the electrochemical impedance loop for Cu/HEA is greater than that for uncoated copper and points to better anti-corrosion properties of the HEA-modified copper electrode compared to the unprotected copper. This conclusion can be connected with lines 257-259.

Lines 262-264 and 253-256: The numerical results in Table 4 should be comment!! Anti-corrosion properties of coating-modified metals and alloys can be numerically evaluated by total resistance of the system, by polarization resistance, Rp, which is direct correlated with anti-corrosion properties of the investigated coating. This Rp was also calculated from polarization measurements (Fig. 5).

Rp, from EIS measuremts, can be calculated as: Rp = Rct+Rf (for bare copper) and Rp = Rct+Rc (for HEA-modified copper). From Rp values, anti-corrosion efficiency can be calculated as :

Efficiency = [(Rpcopper – Rp modified copper)/ Rpcopper]*100%.

Since the phase angle is a structurally sensitive parameter, according to the obtained results (phase angle vs. logf) it is clear that the values of the phase angle are low (they should be higher than 80 ° for barrier coating) and indicate the presence of microdefects (pores or cracks) in the coating structure.

These defects can significantly affect the anti-corrosion properties of the coating, since the tested medium is very aggressive with a high concentration of Cl ions that can disrupt the structure of the coating and significantly reduce its barrier properties. Of course this will happen over a longer period of time.

Therefore, it would be good to add 1-2 sentences that the coating can potentially serve as protection for copper in seawater, but the actual anti-corrosion properties need to be investigated over a longer period of time.

Author Response

(The authors gave the same response as above.)

Round 4

Reviewer 2 Report

The following corrections need to be made:

-Figure 8. presents only the Bode diagrams. The Bode diagram is: log IZI vs. log f and -phase angle vs. log f. The Nyquist diagram is Zim vs. Zreal. Therefore, the figure caption has to be corrected.

Lines 260-263: "... presents a maximum very well established assigned at a phase angle of approximate 50°for HEA, hence, in this event is indicated the presence of microdefects (microcracks) in the coating structure."                    If there are microdefects in the coating (results confirm their existence), then the coating is not a barrier. Barrier has to reflect phase angle more than -80°. Obtained values are too low (around -50°) to be characteristic for a barrier.

-Lines 263-266: "The Nyquist and Bode diagrams suggest that the film on the surface of the alloy halted the corrosion process and acted as barrier by the charge transfer phenomenon."  This sentence has to be corrected. It is unclear.  Film acts as a barrier by the charge transfer phenomenon?? Additional, this sentence contradicts the previous one (lines 260-263): film like barrier and microdefects???). 

HEA film provides an additional protection to the copper electrode during immersion in artificial seewater (corrosion  efficiency of 72%), but imperfections present in the film structure could influence on corrosion behaviour during long time-immersion in this aggressive electrolyte. The most probably, these imperfections would become wider and aggressive ions/molecules could easily penetrate inside the film and consequently the HEA anti-corrosion protection would become worse and worse.

Author Response

I corrected MS, in accordance with the relevant recommendations and observations of Prof. Rew 2.
